# Spatial variability of soil chemical properties under different land-uses in Northwest Ethiopia

Gizachew Ayalew Tiruneh[1]¤*, Tiringo Yilak Alemayehu[2], Derege Tsegaye Meshesha[3], Eduardo Saldanha Vogelmann[4], José Miguel Reichert[5], Nigussie Haregeweyn[6]

1 Faculty of Agriculture and Environmental Sciences, Department of Natural Resources Management, Debre Tabor University, Debre Tabor, Ethiopia, 2 Faculty of Agriculture and Environmental Sciences, Department of Plant Sciences, Debre Tabor University, Debre Tabor, Ethiopia, 3 Geospatial Data and Technology Center, College of Agriculture and Environmental Sciences, Bahir Dar University, Bahir Dar, Ethiopia, 4 Biological Sciences Institute, Federal University of Rio Grande, São Lourenço do Sul, Brazil, 5 Soils Department, Federal University of Santa Maria (UFSM), Rio Grande do Sul, Brazil, 6 International Platform for Dryland Research and Education, Tottori University, Tottori, Japan

☯ These authors contributed equally to this work.
¤ Current address: Department of Natural Resources Management, College of Agriculture and Environmental Sciences, Bahir Dar University, Bahir Dar, Ethiopia
* tiruneh1972@gmail.com

**Data Availability Statement:** All relevant data are within the manuscript and its Supporting Information files.

## Abstract

The understanding of the spatial variation of soil chemical properties is critical in agriculture and the environment. To assess the spatial variability of soil chemical properties in the Fogera plain, Ethiopia, we used Inverse Distance Weighting (IDW), pair-wise comparisons, descriptive analysis, and principal component analysis (PCA). In 2019, soil samples were collected at topsoil (a soil depth of 0–20 cm) from three representative land-uses (cropland, plantation forestland, and grazing lands) using a grid-sampling design. The variance analysis for soil pH, available phosphorus (avP), organic carbon (OC), total nitrogen (TN), electrical conductivity (EC), exchangeable potassium (exchK), exchangeable calcium (exchCa), and cation exchange capacity (CEC) revealed significant differences among the land-uses. The highest mean values of pH (8.9), avP (32.99 ppm), OC (4.82%), TN (0.39%), EC (2.28 dS m$^{-1}$), and exchK (2.89 cmol (+) kg$^{-1}$) were determined under grazing land. The lowest pH (6.2), OC (2.3%), TN (0.15%), and EC (0.11 dS m$^{-1}$) were recorded in cultivated land. The PCA result revealed that the land-use change was responsible for most soil chemical properties, accounting for 93.32%. Soil maps can help identify the nutrient status, update management options, and increase productivity and profit. The expansion of cultivated lands resulted in a significant decrease in soil organic matter. Thus, soil management strategies should be tailored to replenish the soil nutrient content while maintaining agricultural productivity in the Fogera plain.

**Funding:** This work was supported by the Debre Tabor University and Bahir Dar University. The author GAT received the award. The funders had no role in study design, data collection and analysis, decision to publish, or preparation of the manuscript.

**Competing interests:** The authors have declared that no competing interests exist.

## Introduction

Environmental degradation caused by irrelevant land-use is a global problem in sustainable agriculture. Land-use change markedly affects soil properties [1,2]. Changing land-use from forest cover to plough-land may result in a decrease in soil fertility, nutrients, and thus productivity [3–6], as well as increased soil perturbation [5,7–12].

Rapid population growth and environmental factors in Ethiopia have resulted in converting forestland and grassland to cultivated land [13]. The expansion of cultivated areas has a substantial influence on soil nutrient content [14]. [15] reported changes in the amount of soil organic carbon and total N due to changes in land-use and land-cover in the Gerado catchment, northeastern Ethiopia. [16] also reported that deforestation has led to the deterioration of soil organic matter. As a result, soil nutrient deficiency is a critical problem in the country and a major crop production constraint [17,18].

Ethiopia has seen an increase in cultivated lands and eucalyptus plantations and decreased grazing lands because of population growth [13]. The eucalyptus plantation had a significant impact on soil properties [19–21]. [22] reported a reduction in soil organic carbon (OC), total nitrogen (TN), exchangeable cations, and cation exchange capacity (CEC) owing to the shift from woodlands to croplands and grazing lands in the same country. [23] also found a decline in pH and soil organic matter content in cultivated land in Ethiopia's Kabe watershed. As a result, scientific records of spatial variability and distribution of soil properties among land-use shifts are critical for optimizing fertilizer use and increasing crop productivity [20].

In developing countries, including Ethiopia, land-use/land-cover change is a significant source of greenhouse gases (GHGs) such as carbon dioxide ($CO_2$), nitrous oxide ($N_2O$), and methane ($CH_4$) emissions [24]. Furthermore, nitrogen-containing fertilizers [25,26], tillage [26], and complete removal of vegetation and residues [27] have influenced spatial variability, soil nutrient cycling, and GHGs emission. Thus, information on the spatial variability of soil due to land-use change is critical in this regard.

Accurate and scientific information about soils is essential for developing effective soil management techniques that sustain agricultural production while maintaining environmental quality. Furthermore, site-specific management of pH, organic carbon, available N, available P, and available K [28] improves input use efficiency [29], increases crop production economic returns, and reduces ecological risks [30].

Many researchers have recently used geostatistics to estimate the spatial variability of soil properties [31–34]. In geostatistics, the inverse distance weighting (IDW) model can be used to map the spatial distribution of any soil property measured for spatially distributed samples [35]. Understanding the spatial variability of soil properties [36] and developing site-specific recommendations [36,37] are critical for optimizing nutrient usage, improving crop performance, and minimizing environmental risks [38].

Furthermore, the spatial information produced using geostatistical techniques would be an input to improve food security and obtain sustainable yield in developing countries, including Ethiopia, which has never been thoroughly investigated using spatial prediction models [39]. Understanding of different land's soil fertility of various land-use types could be used to predict, monitor, and evaluate the effects of changes in land-use types on soil properties, scheming appropriate land-use planning, and sustaining agricultural productivity. In this regard, previous researches on Fogera plain area have not yet adequately discussed the Fogera plain area. Thus, the target of this research paper was to assess the effects of land-use types on spatial variability and distribution of soil fertility qualities, such as pH, organic carbon (OC), total nitrogen (TN), available phosphorus (avP), exchangeable calcium ($Ca^{2+}$), exchangeable potassium ($K^+$), electrical conductivity (EC), and cation exchange capacity (CEC) in Fogera plain, the highland of Ethiopia.

## Materials and methods

### Description of the study area

The research was conducted in the Fogera plain area (37˚ 0′ 0″ E - 38˚ 45′ 0″ E and 11˚ 15′ 0″ N-12˚ 15′ 0″ N) in the northwest highland of Amhara region, Ethiopia (Fig 1). It has a total area of 5,646 hectares. The topography of the study area is flat land. Rice (*Oryza sativa* L.), maize (*Zea mays* L.), and *Teff* [*Eragrostis tef* (Zucc.)] are the main crops grown in the study region. Rice production (76%) is primarily a subsistence farming operation in the study area. Crop residues are collected for use as fuelwood or animal feed. As a result, no crop residue remains in the field to serve as a source of organic amendments. Rice, onions, and eucalyptus products are essential sources of income for the local people.

### Sources of spatial data and their extraction

The study area's land-use/land-covers for 2019 was derived from an Ethiopia Mapping Agency (EMA) 1:20,000 scale land-use/land-cover map, and the mainland-use system consisted of

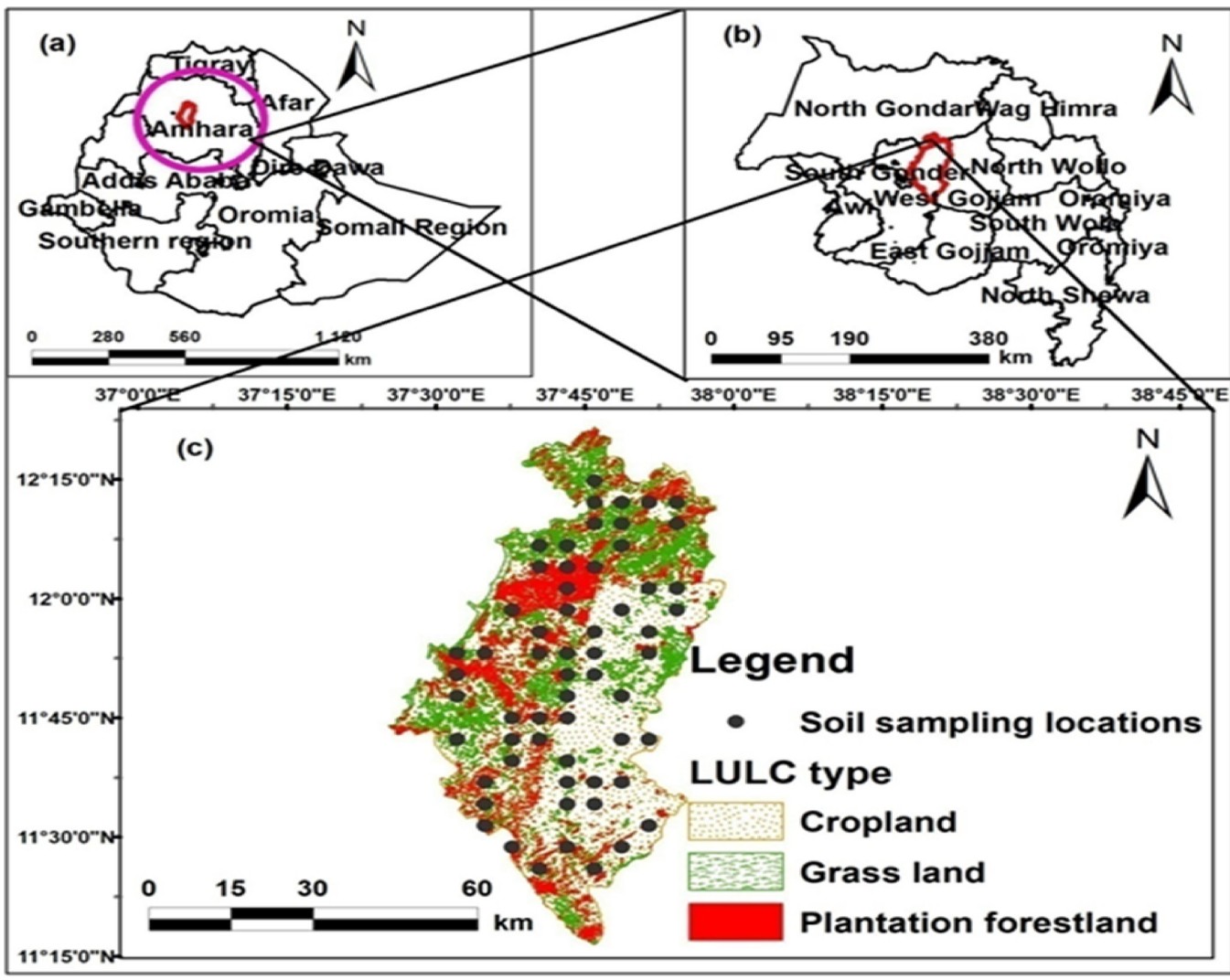

**Fig 1. The study area's location: (a) Ethiopia, (b) Amhara region, and (c) soil sampling.**

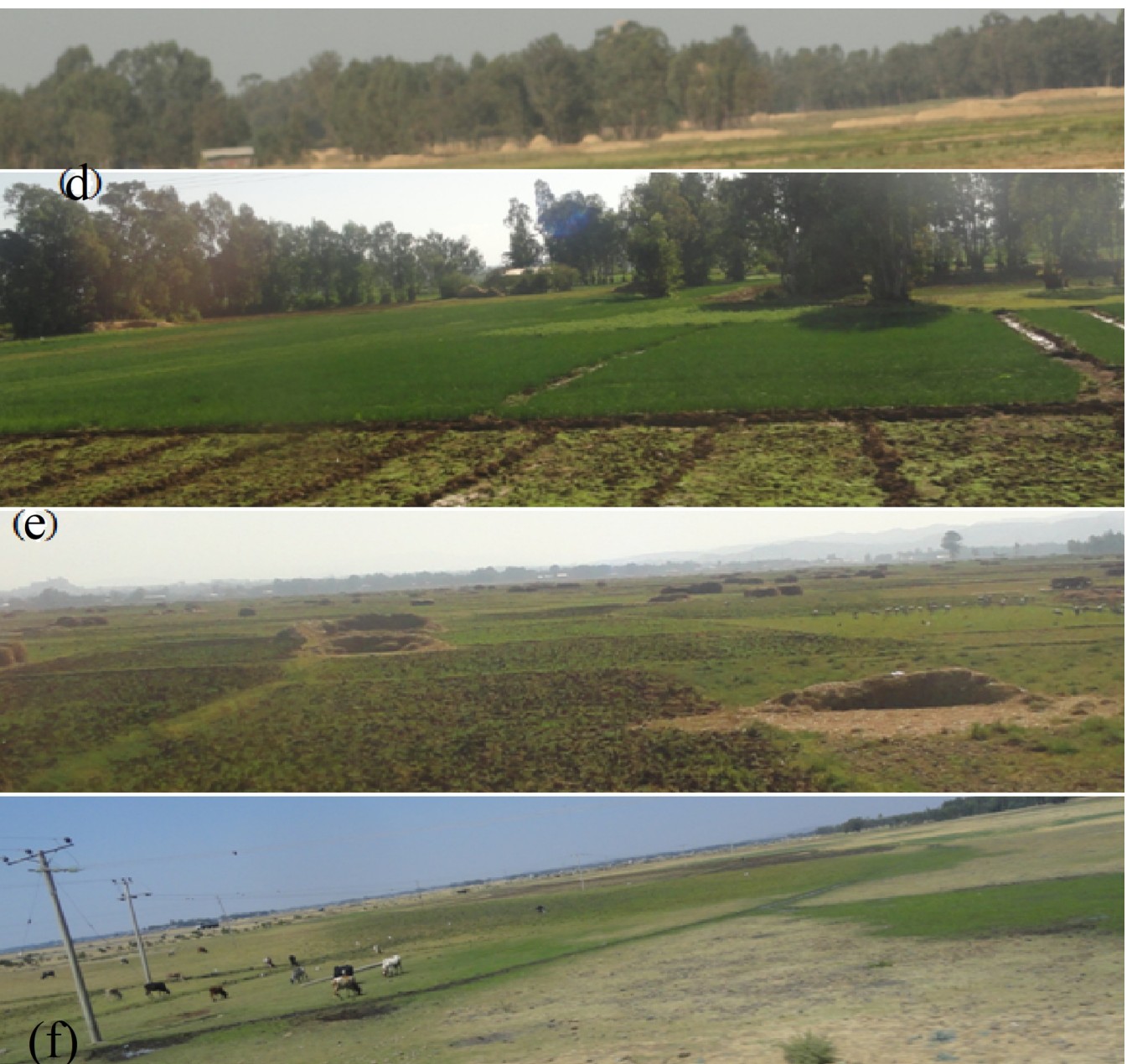

**Fig 2. Land-use of the area: (d) Eucalyptus plantation forestland, (e) Cultivated land, and (f) Grazing land.**

cropland (4,362.73 ha, 77.27%), plantation forestland (6,96.29 ha, 12.33%) and grassland (586.98 ha, 10.39%). We also performed reconnaissance surveys from August to November/ 2019 to validate the map. Using ArcGIS software version 10.5, a digital elevation model (ASTER DEM) with a resolution of 20 * 20 m, downloaded from the EMA website [40], was used to generate elevation and slope of the study area.

As shown Figs 2 and 3, agroecology belonging to Kolla (<1800 m a.s.l.), Weyna dega (1,800–2,400 m a.s.l.), and dega (>2400 m a.s.l.) has an area share of 16.24%, 71.93%, and 11.83%, respectively. According to the digital soil map obtained from Water and Land Resource Centre (WLRC), the soil types in the Fogera plain region are Chromic Vertisols

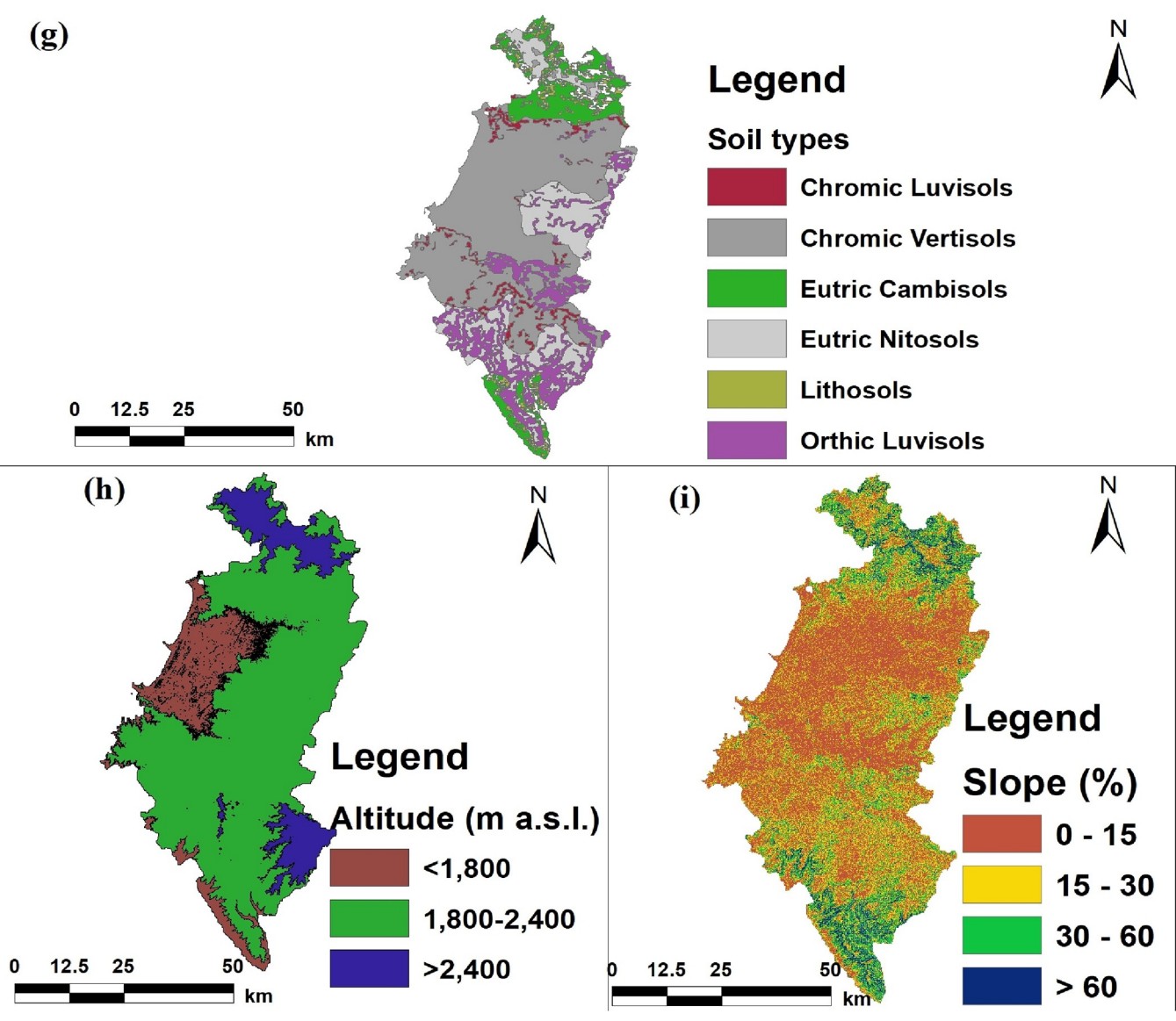

**Fig 3. Different maps of the study area, particularly (g) Soil types, (h) Altitude, and (i) Slope.**

(48.57%), Eutric Nitosols (17.65%), Orthic Luvisols (15.75%), Eutric Cambisols (10.78%), Chromic Luvisols (3.97%), and Lithosols (3.27%) [41].

## Soil sampling, laboratory analysis, and statistical analysis

We used the fishnet tool included with ArcGIS software version 10.5 to build a regularly spaced grid of sampling points on the land-uses in the study region. Following that, 60 representative soil samples (20 from each land use: cropland, plantation forestland, and grazing land) were identified at topsoil (a depth of 0–20 cm) in February-April/2020 using a systematic purposive approach. The topsoil was chosen as plants and soil management practices most influence it. To record each grid center in the field and the latitudes, longitudes, and elevations of sampling points, a portable Global Positioning System (Garmin 60; 2 m accuracy) was used. Soil sampling locations were chosen to reflect each land-use condition by taking topographic

features and soil conditions into account [42]. Each soil sample was created by compositing five sub-samples, improving sampling intensity and lowering soil analysis costs [43,44]. A kilo-gram of soil sample was collected from each location, air-dried, ground using a mortar and pestle, and analyzed at the Amhara Regional Soil Laboratory Center following national stan-dard research methods [45].

Soil pH was measured potentiometrically in $H_2O$ at the soil to solution ratio (1: 2.5) using a combined glass electrode pH meter (Model CP-505, Zabrze ul, Poland) [46]. The electrical conductivity (EC) of the soil was determined using an EC meter at the soil to water ratio of 1: 5 (Orion Model 145, USA) [47]. The Walkley-Black procedures were used to measure soil organic carbon (OC). A weighed portion (1–2 gm) of the dried, ground soil samples were treated with 5 ml of 0.4 N potassium dichromate solution ($K_2Cr_2O_7$) followed by the addition of 10 ml of concentrated sulfuric acid. The mixture was gently mixed and left for 16–18 hours before being given 100 ml of triple-distilled water. The excess of dichromate was back-titrated with the standard 0.2 N ferrous ammonium sulfate solution. The acidic dichromate was blankly titrated with ferrous ammonium sulfate solution [45,48].

The total nitrogen (TN) of the soils was determined through digestion, distillation, and titration procedures of the Kjeldahl using the Kjeldahl apparatus (Gallenhamp, USA) [49]. The soil's available phosphorus (avP) was measured using 0.5 M $NaHCO_3$, pH of 8.5, a soil to solu-tion ratio of 1: 20 for half an hour. The (avP) was extracted with 1 M ammonium chloride, 0.5 M ammonium fluoride, 0.1 M sodium hydro-oxide (from Blulux Laboratory Reagent (p) Ltd), and the amount was measured using a spectrophotometer (UV1700, Japan) [50,51]. Exchange-able calcium (Ca), magnesium (Mg), potassium (K), and sodium (Na) were determined by sat-urating the soil samples with 1 M ammonium acetate solution at pH 7.0. Subsequently, Ca and Mg were determined using Perkin-Elmer Model 290 atomic absorption spectrophotometer (ColVisTec, Germany); while exchangeable Na and K were measured using a Model 18 Per-kin-Elmer flame photometer [52].

The soil's cation exchange capacity (CEC) was calculated by replacing $NH_4^+$ saturated sam-ples with $K^+$ from a percolated KCl solution (from LOBA CHEMIE PVT.LTD). Washing with ethanol (from LOBA CHEMIE PVT.LTD) eliminated excess salt, and $NH_4^+$ was displaced by $K^+$ [53]. Merck KGaA and Sigma-Aldrich, Steinheim, Germany supplied all of the chemicals and reagents, such as potassium dichromate, sulphuric acid, ferrous sulfate, ferroin, sodium bicarbonate, ammonium chloride, ammonium fluoride, and ammonium acetate, unless other-wise mentioned.

The variation of soil organic carbon, total nitrogen, and potassium was defined using the Inverse Distance Weighting (IDW) model [54,55]. Furthermore, IDW has been widely used by scholars for the prediction of soil OM and soil nitrate [56], P and K levels [57], and soil pH scale [58]. The current study used IDW to map the spatial distribution of the soil chemi-cal properties under the ArcGIS environment. Besides, pair-wise comparisons, descriptive analysis, and principal component analysis (PCA) were performed using Statistical Analyses System (SAS) software version 9.4 and the Statistical Package for Social Sciences (SPSS) software version 24, respectively. Means were compared through the Tukey test at 1% probability.

## Ethics statement

Debre Tabor University's Research and Publication Directorate and Bahir Dar University's Research and Publication Directorate authorized the present study to collect soil samples and access the field site. Farmers agreed to collect the soil samples in the study area, as the survey has no harmful effects on humans.

**Table 1. Soil pH$_{H2O}$ rating, area share (ha, %) by land-use, and parametric test.**

| Common name | pH rating | Area share (ha, %) by land-use | | | | | |
|---|---|---|---|---|---|---|---|
| | | Cropland | | Plantation forestland | | Grazing land | |
| | | Area (ha) | % | Area (ha) | % | Area (ha) | % |
| Moderately acid | 5.6–6.5 | 181.84 | 4.17 | 1.11 | 0.16 | 9.50 | 1.62 |
| Neutral | 6.6–7.3 | 1,576.93 | 36.15 | 139.05 | 19.97 | 173.04 | 29.48 |
| Moderately alkaline | 7.4–8.4 | 2,603.96 | 59.69 | 556.12 | 79.87 | 404.44 | 68.90 |
| Total | | 4,362.73 | 100 | 696.29 | 100 | 586.98 | 100 |
| pH (mean ± standard error) | | 6.58 ± 0.05 [a] | | 7.40 ± 0.05 [b] | | 8.44 ± 0.05 [c] | |

Means of pH with different letters are significantly varied (Tukey, p < 0.01).

## Results and discussion

### Effect of land-use/land-cover types on soil fertility quality

**Soil pH$_{H2O}$.**    The soil pH, which affects nutrient availability, varied significantly (P < 0.05) depending on land-use type (Table 1). The soil pH values were found to be the highest (8.9) and the lowest (6.2) under the grazing and the cultivated lands, respectively (S2 Table and Figs 4 and 5). According to [59], higher soil pH levels obtained from plantation forestland and

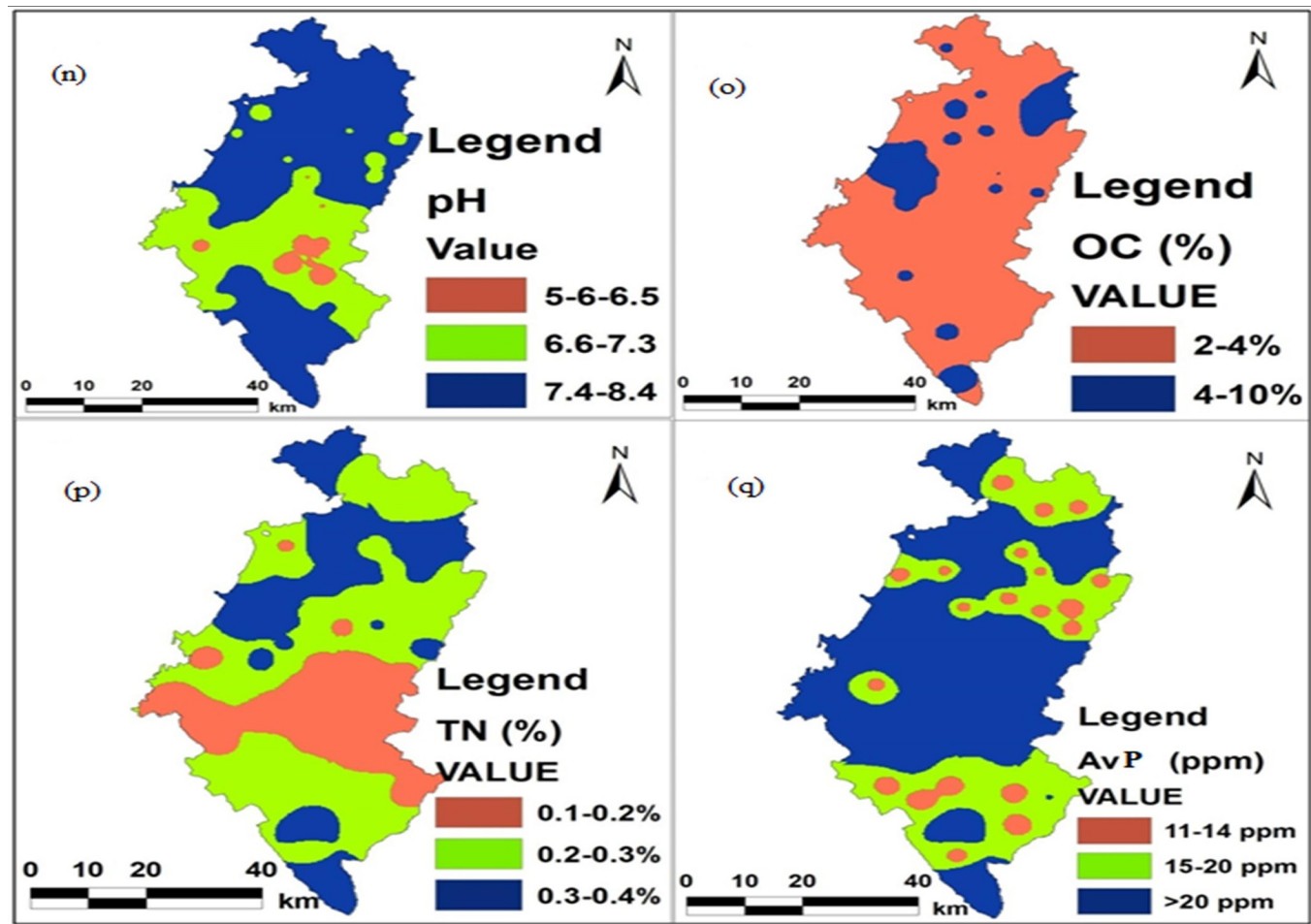

**Fig 4. Maps of (n) Soil pH, (o) Organic carbon, (p) Total nitrogen, and (q) Available phosphorus.**

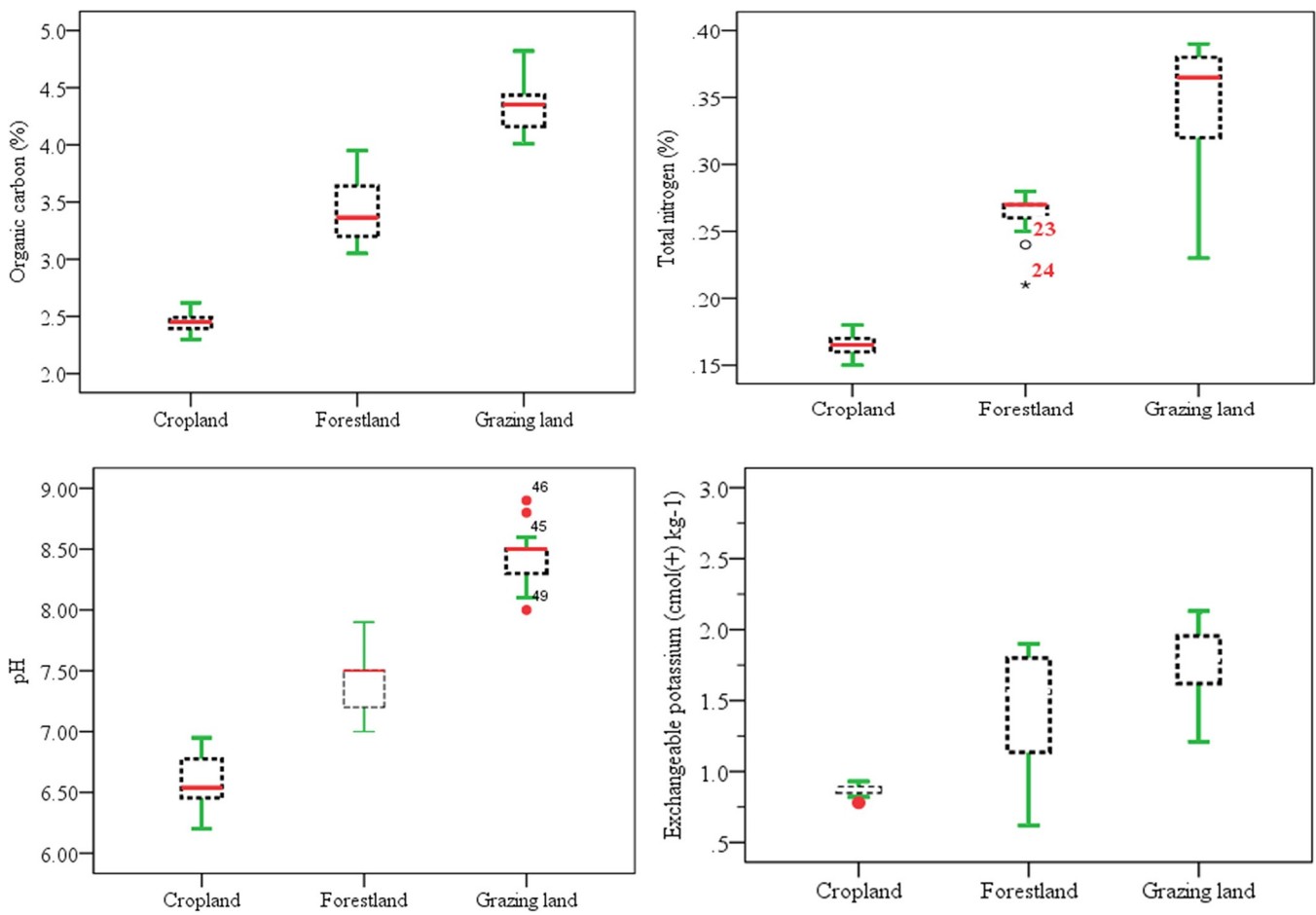

**Fig 5. Mean values of soil parameters using box plots.**

grazing land could be associated with the presence of basic cations emanated from weathering [23] and the potash obtained from ashes [60].

**Available phosphorus (avP).** The analysis of variance showed a significant difference in the mean value of avP among land-use types. Following the limit established by [61], very high (>20 ppm) avP content was observed in soils of all land-uses (Table 2 and Fig 4). Higher avP values in soils could be revealed by the recurrent use of mineralized phosphorus [38], and the

**Table 2. Soil available phosphorus (avP) rating, area share (ha, %) by land-use, and parametric test.**

| Common name | AvPrate | Area share (%) by land-use | | | | | |
|---|---|---|---|---|---|---|---|
| | | Cropland | | Plantation forestland | | Grazing land | |
| | ppm | Area (ha) | Area (%) | Area (ha) | Area (%) | Area (ha) | Area (%) |
| Medium | 11–14 | 273.66 | 6.27 | 35.02 | 5.03 | 23.05 | 3.93 |
| High | 15–20 | 1,425.74 | 32.68 | 200.90 | 28.85 | 118.12 | 20.12 |
| Very high | >20 | 2,663.33 | 61.05 | 460.37 | 66.12 | 445.82 | 75.95 |
| Total | | 4,362.73 | 100 | 696.29 | 100 | 586.98 | 100 |
| AvP (mean ± standard error) | | 21.32 ± 0.12 [a] | | 11.43± 0.06 [b] | | 32.52± 0.06 [c] | |

Means of available P with different letters are significantly different (Tukey, p < 0.01).

**Table 3. Soil organic carbon (OC) rating, area share (ha, %) by land-use, and parametric test.**

| Common name | OC rate | Area share (ha, %) by land-use | | | | | |
|---|---|---|---|---|---|---|---|
| | | Cropland | | Plantation forestland | | Grazing land | |
| | % | Area (ha) | Area (%) | Area (ha) | Area (%) | Area (ha) | Area (%) |
| Low | 2–4 | 3,824.5 | 87.66 | 561.44 | 80.63 | 479.59 | 81.70 |
| Medium | 4–8 | 538.23 | 12.34 | 134.85 | 19.37 | 107.40 | 18.30 |
| Total | | 4,362.73 | 100 | 696.29 | 100 | 586.98 | 100 |
| OC (mean ± standard error) | | 2.44 ± 0.02 [a] | | 3.43 ± 0.06 [b] | | 4.35± 0.06 [c] | |

Means of soil OC with different letters are significantly different (Tukey, p < 0.01).

addition of manure, compost, and ashes [62], presence of weathered soil minerals [63], and actions of microbes. Higher levels of avP in the soils indicate that the soils have optimum nutrients for crop growth. [22,64] reported similar findings in Ethiopia. However, regular monitoring of the availability of phosphorus in the soil is essential.

**Soil organic carbon (OC) and total nitrogen (TN).** Land-use changes caused a significant difference in soil organic carbon (OC) and total nitrogen (TN). According to the ranking set by [65], low organic carbon content (2.44%) in the soils dominated the agricultural land (87.66%), grazing land (81.7%), and eucalyptus plantation forestland (80.63%), as shown in Table 3 and Figs 4 and 5. The highest (4.35%) and lowest (2.44%) OC contents obtained in grazing and croplands demonstrated that soil OC showed a better response to land-use type. The variations in the mean value of soil organic carbon and total nitrogen could have attributed to high erosion rates, crop residue exclusion, increased mineralization rates, and nutrient deficiency [38,66]. The higher organic carbon and available P contents of the grazing lands suggest that OM is the primary source of avP [67].

According to the rate of [61], the cropland and plantation forestland demonstrated, respectively, low (0.17%) and high total nitrogen (0.35%) contents in the soils (Table 4 and Figs 4 and 5). In line with this, croplands had lower soil OC content [2,23,68]. Higher soil OC and TN contents found in plantation forests and grazing lands are most likely due to grass burning and dung deposition, respectively. Furthermore, researchers have advocated for grazing to sustain nutrient cycling and decomposition rates [69]. The low total nitrogen content may be eligible to minimize nitrogen loss by volatilization or leaching and rapid decomposition of OM. Hence, grassland and eucalyptus plantation conversion to cultivated land worsens soil OC and TN decline [70]. Thus, for long-term development, the soils need external nitrogen and carbon inputs.

**Table 4. Soil total nitrogen (TN) rating, area share (ha, %) by land-use, and parametric test.**

| Common name | TN rate | Area share (ha, %) by land-use | | | | | |
|---|---|---|---|---|---|---|---|
| | | Cropland | | Plantation forestland | | Grazing land | |
| | % | Area (ha) | Area (%) | Area (ha) | Area (%) | Area (ha) | Area (%) |
| Low | 0.1–0.2 | 1,219.37 | 27.95 | 69.66 | 16.29 | 115.14 | 19.62 |
| Medium | 0.2–0.3 | 2,159.42 | 49.50 | 62.31 | 14.57 | 244.65 | 41.68 |
| High | 0.3–0.4 | 983.94 | 22.55 | 295.63 | 69.14 | 227.18 | 38.70 |
| Total | | 4,362.73 | 100 | 427.60 | 100 | 586.98 | 100 |
| TN (mean ± standard error) | | 0.17 ± 0.0 [a] | | 0.27 ± 0.01 [b] | | 0.35± 0.0 [c] | |

Means of TN with different letters are significantly different (Tukey, p < 0.01).

**Table 5. Soil exchangeable potassium (exchK) rating, area share (ha, %) by land-use, and parametric test.**

| Common name | ExchK (rate) | Area share (ha, %) by land-use | | | | | |
| --- | --- | --- | --- | --- | --- | --- | --- |
| | | Cropland | | Plantation forestland | | Grazing land | |
| | $cmol_{(+)}$ $kg^{-1}$ | Area (ha) | Area (%) | Area (%) | Area (ha) | Area (%) | Area (ha) |
| High | 0.51–1.51 | 2,609.17 | 59.81 | 320.17 | 45.98 | 257.26 | 43.83 |
| Medium | 1.51–2.3 | 1,733.37 | 39.73 | 373.31 | 53.61 | 326.57 | 55.64 |
| Very high | >2.3 | 20.19 | 0.46 | 2.80 | 0.40 | 3.15 | 0.54 |
| Total | | 4,362.73 | 100 | 696.29 | 100 | 586.98 | 100 |
| ExchK (mean ± standard error) | | 0.87± 0.01 [a] | | 1.47 ± 0.08 [b] | | 1.81 ± 0.08 [c] | |

Means of exchK with different letters are significantly different (Tukey, $p < 0.01$).

**Exchangeable cations (K, Ca) and electrical conductivity (EC).** Next to nitrogen and phosphorus, potassium is the third most important essential element that limits crop productivity. As shown in Tables 5 and 6 and Figs 5 and 6, there was a significant variation in soil K, Ca, and EC contents based on land-use. Besides, according to [59,61], the soils in the study region had high Ca (>20 $cmol_{(+)}$ $kg^{-1}$) and K (0.51–1.51 $cmol_{(+)}$ $kg^{-1}$) contents. In the soils, higher Ca and K levels were found. It might be due to the type of parent materials, weathering, land-use types, fertilizer types, and leaching rates, crop remains, and litter fall [71]. The higher Ca and K contents present in grazing and plantation forestlands are associated with the higher pH value [72] and clay particles [73]. On the other hand, the soils showed slightly salty.

This result suggests that Ca and K do not appear to be limiting nutrients to crop production in the region. Based on the EC rate established by [74], no significant amounts of soluble salts were accumulated, implying that plant growth and development would be unaffected.

**Cation Exchange Capacity (CEC).** According to the rate set by [61], a significant variance and higher CEC indicated that soils in the study area have a high capacity to retain nutrients against leaching losses (Table 7 and Fig 6). The highest CEC values recorded in cultivated land-use may be soil organic material, pH, quantity, and type of clay, which adsorb and retain positive cations through electrostatic force [75]. The current findings were also consistent with [76–78], who reported higher CEC under cultivated lands in Ethiopia's highlands.

## Principal component analysis (PCA) of soil chemical properties

The first two principal component analyses (PCA) with eigenvalues greater than one were able to explain the most significant variance (93.32%) of the analyzed soil chemical properties (S1 Fig) [79]. Moreover, 76.77% of the variation in data was explained by pH, organic carbon

**Table 6. Electrical conductivity (EC) rating, area share (ha, %) by land-use, and parametric test.**

| Common name | EC rate | Area share (ha, %) by land-use | | | | | |
| --- | --- | --- | --- | --- | --- | --- | --- |
| | | Cropland | | Plantation forestland | | Grazing land | |
| | dS $m^{-1}$ | Area (ha) | Area (%) | Area (ha) | Area (%) | Area (ha) | Area (%) |
| Not salty | <0.75 | 1,427.14 | 32.71 | 96.07 | 13.80 | 146.18 | 24.90 |
| Slightly salty | 0.75–2 | 2,660.19 | 60.98 | 538.15 | 77.29 | 372.90 | 63.53 |
| Moderately salty | 2–4 | 275.30 | 6.31 | 62.07 | 8.91 | 67.90 | 11.57 |
| Sum | | 4,362.63 | 100 | 696.29 | 100 | 586.98 | 100 |
| EC (mean ± standard error) | | 0.13± 0.0 [a] | | 1.21 ± 0.01 [b] | | 2.22± 0.01 [c] | |

Means of EC with different letters are significantly different (Tukey, $p < 0.01$).

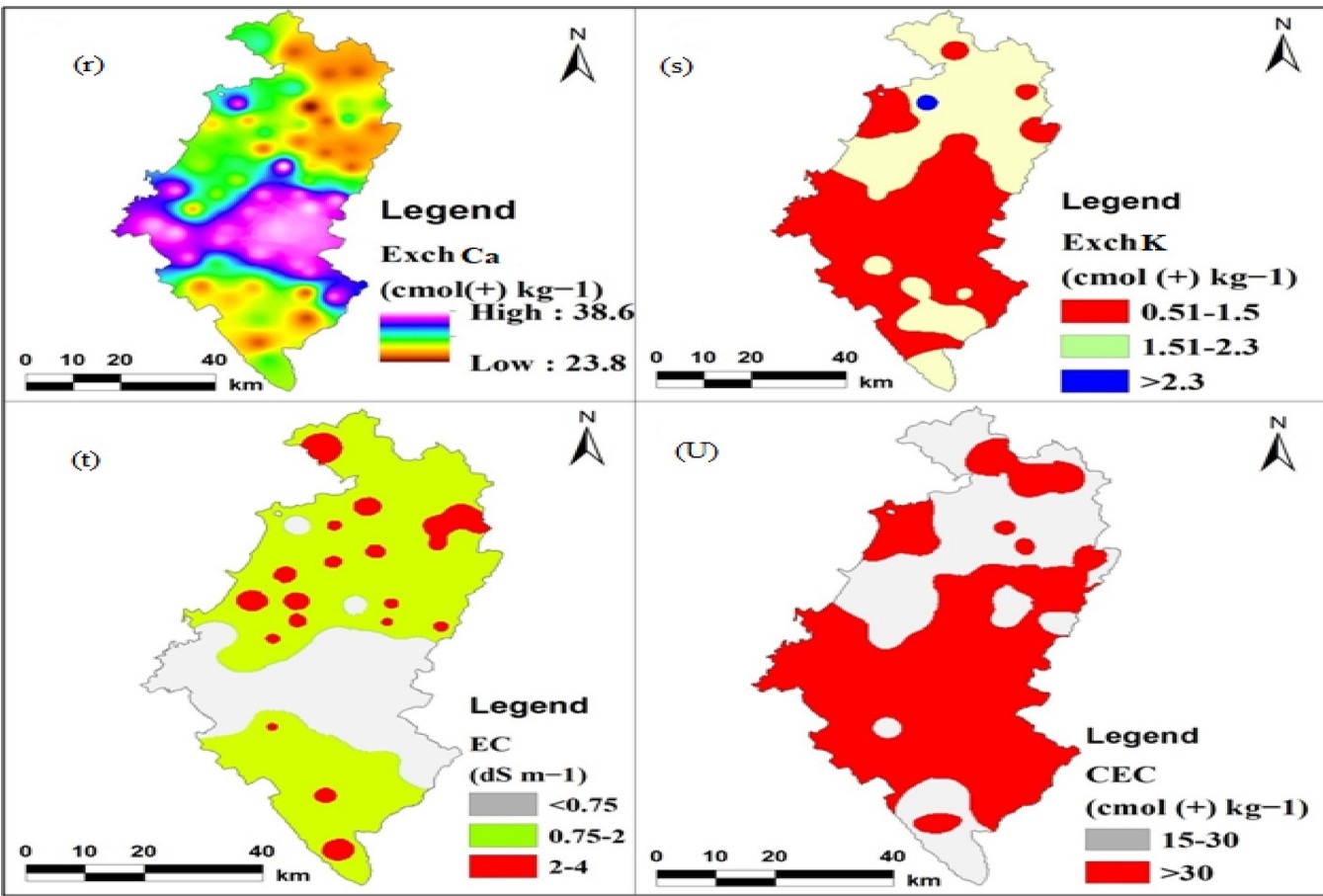

**Fig 6. Maps: (r) Exchangeable calcium, (s) Exchangeable potassium, (t) Electrical conductivity, and (u) Cation exchange capacity.**

(OC), total nitrogen (TN), exchangeable potassium (exchK), electrical conductivity (EC), and cation exchange capacity (CEC) on the first PC. Simultaneously, the second component notably loaded the available phosphorus (avP) and exchangeable calcium (exchCa) (Table 8 and S1 Fig). The communality is the proportion of the variation of a variable retained in a component. The communalities of the two PCs explained by each soil variable ranged from 73 to 99%. In PC 1, CEC showed a higher loading value (-1.0) and communality (99%) and is called 'cation exchange capacity, CEC factor.' While in PC 2, available phosphorus showed a higher loading

**Table 7. Cation exchange capacity (CEC) rating, area share (ha, %) by land-use, and parametric test.**

| Common name | CEC (rate) | Area share (ha, %) by land-use | | | | | |
| --- | --- | --- | --- | --- | --- | --- | --- |
| | | Cultivated land | | Plantation forestland | | Grazing land | |
| | $cmol_{(+)}\ kg^{-1}$ | Area (ha) | Area (%) | Area (ha) | Area (%) | Area (ha) | Area (%) |
| High | 15–30 | 1,402.31 | 32.14 | 357.07 | 51.28 | 275.72 | 46.97 |
| Very high | >30 | 2,960.42 | 67.86 | 339.22 | 48.72 | 311.26 | 53.03 |
| Sum | | 4,362.73 | 100 | 696.29 | 100 | 586.98 | 100 |
| CEC (mean ± standard error) | | 40.36 ± 0.10 [a] | | 31.53 ± 0.06 [b] | | 25.24 ± 0.06 [c] | |

Means of CEC with different letters are significantly different (Tukey, p < 0.01).

**Table 8. Principal component analysis of soil chemical properties about land-uses in Fogera plain, northwest Ethiopia.**

| Principal component | PC1 | PC2 | |
|---|---|---|---|
| Eigenvalue | 6.07 | 1.29 | |
| Variance (%) | 79.97 | 18.58 | |
| Variables | Eigenvectors | | Communalities |
| pH | **0.96** | 0.12 | 0.94 |
| Available phosphorus | 0.48 | **0.86** | 0.98 |
| Organic carbon | **0.96** | 0.03 | 0.98 |
| Total nitrogen | **0.96** | 0.00 | 0.73 |
| Exchangeable potassium | **0.81** | -0.08 | 0.98 |
| Exchangeable calcium | -0.67 | **0.73** | 0.93 |
| Electrical conductivity | **0.99** | 0.03 | 0.93 |
| Cation exchange capacity | **-1.00** | 0.04 | 0.99 |

Bold eigenvector values referred to highly weighted variables in the PC.

value (0.86) and communality (98%) and termed 'available phosphorus, avP factor.' It indicates that the PCA reduces the dimensions and complexity of the soil data matrix [80].

## Implications for sustainable soil fertility management and environmental conservation

Our results showed that the transition from grassland to cultivated land and eucalyptus plantation significantly reduced the total nitrogen within Fogera plain's topsoil. Organic materials may increase the nitrogen content in the soil, which has a more significant effect on crop growth and yield than other nutrients. Nevertheless, avP range of the area's soils was high (>20 ppm), which could be attributed to the frequent use of mineralized phosphorus [38]. The addition of nitrogen-containing fertilizer inputs might also improve the cultivated lands' soil nutrient supply to better crop yield and farming profitability. However, the overuse of nitrogen and phosphorus fertilizer may lead to global climate change due to their energy-intensive processing and inefficient use [81], eutrophication of water bodies [82], and soil acidification [83]. Besides, total greenhouse gas emissions, including $N_2O$, $CO_2$ have increased under cultivated lands, depending on the decomposition of organic materials in the soil [81]. Soil management methods, optimum N application rate [84], organic resources, and nitrification inhibitors are all possible soil management approaches.

The spatial soil variability across land-uses is vital for sustainable land management practices, reducing soil erosion, enhancing land productivity, improving farmers' livelihood, reducing GHGs, and maintaining environmental quality [85,86]. Furthermore, we should develop relevant land-use planning and policies to provide an optimal solution geared toward improving the soil's nutrient use efficiency and reducing the adverse environmental effects, including nitrate losses to water and $N_2O$ emissions [87,88].

## Conclusion

The soil pH, available phosphorus (avP), organic carbon (OC), total nitrogen (TN), electrical conductivity (EC), exchangeable bases (Ca and K), and cation exchange capacity (CEC) were varied among land-use types in the Fogera plain. Grazing land had the highest values of pH (8.9), avP (32.99 ppm), OC (4.82%), TN (0.39%), EC (2.28 dS m$^{-1}$), and exchK (2.89 cmol$_{(+)}$ kg$^{-1}$), while in cultivated land had the lowest OC (2.3%), TN (0.15%), soil pH (6.2), and EC

(0.11 dS m$^{-1}$). The difference in the land-use types could be associated with the variation of soil chemical properties in the study area.

The study found that the expansion of cultivated lands depleted soil OC and TN, restricting crop growth and decreasing crop yield. Thus, proper nutrient management strategies, such as adding organic and inorganic materials, should be adopted, especially for these nutrients. Besides, PCA prioritized the CEC and avP as the most critical soil chemical properties across land-use types in the study area. Priority should also be given to these selected variables as they provide reliable and on-time information about soil chemical properties and nutrient contents under study area conditions.

Moreover, soil properties' maps improve soil management alternatives, optimize fertilizer use, and enhance crop productivity, thus contributing to the nation's food security. Models should gear to larger samples in future studies to understand better the spatial variability of soil properties of the Fogera plain, Ethiopia.

## Supporting information

**S1 Fig. Loading plot (y) and scree plot (z).** Organic carbon (OC), Total nitrogen (TN) Available phosphorus (avP), Exchangeable calcium (ExchCa), Exchangeable potassium (ExchK), Cation exchange capacity (CEC), and Electrical conductivity (EC).
(TIF)

**S2 Fig. Q-Q plots of soil fertility parameters in Fogera plain.** pH, Cation exchange capacity (CEC), Organic carbon (OC), and Total nitrogen (TN).
(TIF)

**S1 Table. Laboratory results of soil chemical properties in Fogera plain.**
(XLS)

**S2 Table. Descriptive statistics of soil chemical properties in Fogera plain.**
(XLS)

## Acknowledgments

We substantially acknowledged the Amhara Regional Soil Laboratory Center for laboratory analysis for this study. Besides, we acknowledged the Water and Land Resource Centre (WLRC) and Ethiopian Mapping Agency (EMA) for various data access.

## Author Contributions

**Conceptualization:** Gizachew Ayalew Tiruneh, Derege Tsegaye Meshesha, José Miguel Reichert, Nigussie Haregeweyn.

**Data curation:** Derege Tsegaye Meshesha, Eduardo Saldanha Vogelmann, José Miguel Reichert.

**Formal analysis:** Gizachew Ayalew Tiruneh, Tiringo Yilak Alemayehu, José Miguel Reichert.

**Funding acquisition:** Gizachew Ayalew Tiruneh.

**Investigation:** Eduardo Saldanha Vogelmann.

**Methodology:** Tiringo Yilak Alemayehu, Derege Tsegaye Meshesha.

**Project administration:** Derege Tsegaye Meshesha.

**Supervision:** Derege Tsegaye Meshesha, José Miguel Reichert, Nigussie Haregeweyn.

**Validation:** Tiringo Yilak Alemayehu.

**Writing – original draft:** Gizachew Ayalew Tiruneh.

**Writing – review & editing:** Gizachew Ayalew Tiruneh, Tiringo Yilak Alemayehu, Derege Tsegaye Meshesha, Eduardo Saldanha Vogelmann, José Miguel Reichert, Nigussie Haregeweyn.

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
