## [Decision Letter · Decision Letter 0]

26 Mar 2021

PONE-D-21-03433

Spatial variability of soil properties under different land-uses in Northwest
Ethiopia

PLOS ONE

Dear Dr. Tiruneh,

Thank you for submitting your manuscript to PLOS ONE. After careful consideration, we
feel that it has merit but does not fully meet PLOS ONE’s publication criteria as it
currently stands. Therefore, we invite you to submit a revised version of the
manuscript that addresses the points raised during the review process.

Please submit your revised manuscript by May 10 2021 11:59PM. If you will need more
time than this to complete your revisions, please reply to this message or contact
the journal office at plosone@plos.org. When
you're ready to submit your revision, log on to https://www.editorialmanager.com/pone/ and select the 'Submissions
Needing Revision' folder to locate your manuscript file.

If you would like to make changes to your financial disclosure, please include your
updated statement in your cover letter. Guidelines for resubmitting your figure
files are available below the reviewer comments at the end of this letter.

We look forward to receiving your revised manuscript.

Kind regards,

Remigio Paradelo Núñez

Academic Editor

PLOS ONE

Journal Requirements:

2. Please include a separate caption for each figure in your manuscript.

3. Please ensure that you refer to Figures 5 and 6 in your text as, if accepted,
production will need this reference to link the reader to the figure.

Additional Editor Comments (if provided):

Reviewers' comments:

Reviewer's Responses to Questions

**Comments to the Author**

1. Is the manuscript technically sound, and do the data support the conclusions?

Reviewer #1: No

Reviewer #2: Partly

2. Has the statistical analysis been performed
appropriately and rigorously? 

Reviewer #1: No

Reviewer #2: Yes

3. Have the authors made all data underlying the
findings in their manuscript fully available?

Reviewer #1: Yes

Reviewer #2: Yes

4. Is the manuscript presented in an intelligible
fashion and written in standard English?

Reviewer #1: No

Reviewer #2: Yes

5. Review Comments to the Author

Reviewer #1: The manuscript “Spatial variability of soil properties under different
land-uses in Northwest Ethiopia" is well written. In my opinion, the manuscript is
relevant and appropriate for the PlosONE journal. Nonetheless, I have the following
suggestions that should be addressed by the authors before to publish:

Title: please add ‘chemical’ in title

[Not a comment, but you have not defined paragraph sign meaning in author name list-
double check]

L#21-23: these are not only procedure to evaluate soil variability, so please make a
general sentence but focus as background information or rationale of the study

L#24: Include depth of topsoil

Abstract is poorly written. No clear explanation of measured variables. What was the
tool(s) used to explore spatial variability of study area? Only descriptive
statistics?

L#38: what do you mean by inappropriate?

L#43: please double check your citation style for [14] based on journal
guidelines

L#58-61: Besides scholars, your study should be valuable to farmers as well. Please
think about other site-specific recommendation practices and mention it in
Introduction section because there is not only geostatistics for optimizing nutrient
use and improve crop production. Countries like, Nigeria, India, Nepal have used
soil testing mobile van as well. I am not sure about similar approach in Ethiopia,
but it is suggested to visit this link (https://doi.org/10.1080/15427528.2017.1387837) for more
information.

L#111: Is this correct amount? 20-50 mg of dried soil; please cross-check with this
information: https://www.environment.nsw.gov.au/resources/soils/testmethods/oc.pdf

L#136-139: why there is not any information about tools of spatial interpolation in
Introduction section? You need to re-structure your Introduction and Methodology
(statistical analysis) sections.

No justification- why did you select IDW only, why not others?

Results: there is repetition of information (results) in table and text many times.
Also there is no any value of Table 8, just include mean and SE, and then lsd
comparison. If you have this already, then delete Table 8.

L#251-252: How GHG information relates to your current study? Do not bring larger
picture in your Discussion section. Please focus on that what you have done and its
implications.

L#255: what is best land planning?

Good luck!

Reviewer #2: The authors presented a work on "Spatial variability of soil properties
under different land-uses in Northwest Ethiopia" but however dwelt so much on land
use instead of the spatial variability. The authors use only 60 samples to produce
the map of the distribution of soil chemical properties in the study location of
more than 6,000 ha. This to me is not acceptable because the samples are too few to
highlight the spatial variability in the soil properties. Even the map produced was
not mentioned in the abstract let alone discussing them in the results and
Discussion section. A number of information that is supposed to be in the abstract
are missing. There were a lot of long sentences that need to be summarized or broken
into two or more sentences.

6. PLOS authors have the option to publish the peer
review history of their article (what does this mean?). If published, this will
include your full peer review and any attached files.

If you choose “no”, your identity will remain anonymous but your review may still be
made public.

**Do you want your identity to be public for this peer review?** For
information about this choice, including consent withdrawal, please see our
Privacy Policy.

Reviewer #1: No

Reviewer #2: No

---

## [Author Response · Author response to Decision Letter 0]

27 Apr 2021

Date: April 24 09, 2021

Rebuttal letter

PONE-D-21-03433

We are glad about the academic editor and the reviewers’ comments, which strengthen
the current version of the manuscript “Spatial variability of soil properties under
different land-uses in Northwest Ethiopia”. In addition, our utmost sincere
gratitude goes to you and the reviewers who devote their valuable time to bring our
manuscript to a qualified paper. 

We have provided a one by one reply to all concerns and comments given below. We
thank you for your consideration of this resubmission and look forward to your
response.

Best regards,

Gizachew Ayalew Tiruneh (on behalf of all co-authors)

Lecturer in Debre Tabor University

Ph.D. Fellow in soil science, Bahir Dar University

Email: tiruneh1972@gmail.com

 

Editor’s comments

Comments 1: A rebuttal letter that responds to each point raised by the academic
editor and reviewer(s). You should upload this letter as a separate file labeled
'Response to Reviewers'.

Response: We addressed the concerns provided by the editor and reviewers and uploaded
a file labeled “Response to Reviewers”.

Comments 2: A marked-up copy of your manuscript that highlights changes made to the
original version. You should upload this as a separate file labeled 'Revised
Manuscript with Track Changes'.

Response: We tried to do it.

Comments 3: An unmarked version of your revised paper without tracked changes. You
should upload this as a separate file labeled 'Manuscript'.

Response: We have addressed accordingly.

Comments 4: If you would like to make changes to your financial disclosure, please
include your updated statement in your cover letter. 

Response: We have not made any changes to financial disclosure.

Comments 5: Guidelines for resubmitting your figure files are available below the
reviewer comments at the end of this letter.

Response: We made corrections as per the guidelines 

Comments 6: If applicable, we recommend that you deposit your laboratory protocols in
protocols.io to enhance the reproducibility of your results. Protocols.io assigns
your protocol its own identifier (DOI) so that it can be cited independently in the
future. For instructions see: http://journals.plos.org/plosone/s/submission-guidelines#loc-laboratory-protocols.
Additionally, PLOS ONE offers an option for publishing peer-reviewed Lab Protocol
articles, which describe protocols hosted on protocols.io. Read more information on
sharing protocols at https://plos.org/protocols?utm_medium=editorial-email&utm_source=authorletters&utm_campaign=protocols.

Response: The majority of our protocols involve standard methods such as soil pH,
available phosphorus (av. P), organic carbon (OC), total nitrogen (TN), electrical
conductivity (EC), exchangeable Ca, and K, and cation exchange capacity (CEC)
measurement. We have specified our Lab protocol in the revised manuscript [reference
#45]. 

Comments 7: When submitting your revision, we need you to address these additional
requirements, please ensure that your manuscript meets PLOS ONE's style
requirements, including those for file naming. The PLOS ONE style templates can be
found at

Response: We addressed PLOS ONE's style requirements in this revision.

Comments 8: Please include a separate caption for each figure in your manuscript.

Response: We addressed PLOS ONE's style requirements in this revision using PACE.

Comments 9: Please ensure that you refer to Figures 5 and 6 in your text as, if
accepted, production will need this reference to link the reader to the figure.

Response: We cited the figures in manuscript’s text.

Comments 10: Please include captions for your Supporting Information files at the end
of your manuscript, and update any in-text citations to match accordingly. 

Response: We followed PLOS ONE's Supporting Information guidelines to include the
captions.

Reviewers' comments:

Reviewer #1: 

Comments 1: 1. Is the manuscript technically sound, and do the data support the
conclusions?

Reviewer #1: No

Reviewer #2: Partly

Response: Dear Reviewers, thank you so much for taking your valuable time to elevate
the quality of our manuscript. We do hope that the Reviewer’s concerns will be
addressed.

2. Has the statistical analysis been performed appropriately and rigorously?

Reviewer #1: No

Reviewer #2: Yes

Response: Thank you. We have gone thoroughly the revised manuscript, and hopefully
that the first Reviewer will be satisfied.

3. Have the authors made all data underlying the findings in their manuscript fully
available?

Reviewer #1: Yes

Reviewer #2: Yes

Thank you.

4. Is the manuscript presented in an intelligible fashion and written in standard
English?

Reviewer #1: No

Reviewer #2: Yes

Response: Thank you. We have thoroughly revised our manuscript with the help of
Grammarly (premium) and Turnitin software, and we do hope that the first Reviewer’s
concerns will be addressed.

5. Review Comments to the Author

Reviewer #1: 

The manuscript “Spatial variability of soil properties under different land-uses in
Northwest Ethiopia" is well written. In my opinion, the manuscript is relevant and
appropriate for the PlosONE journal. Nonetheless, I have the following suggestions
that should be addressed by the authors before to publish:

Title: please add ‘chemical’ in title

 [Not a comment, but you have not defined paragraph sign meaning in author name list-
double check]

Response: Thank you. We have tried to add ‘chemical’ the title in the way this
reviewer has suggested. 

L#21-23: these are not only procedure to evaluate soil variability, so please make a
general sentence but focus as background information or rationale of the study

Response: Thank you. We have revised the background information or rationale of the
study.

L#24: Include depth of topsoil

Abstract is poorly written. No clear explanation of measured variables. What was the
tool(s) used to explore spatial variability of study area? Only descriptive
statistics?

Response: Thank you. We have included depth of topsoil (0-20 cm) and have revised the
Abstract section. We also incorporated more ideas on geo-statistical (IDW) tool and
others (ANOVA, box plots, and PCA). We hope that this revised version will be
satisfying.

L#38: what do you mean by inappropriate?

Response: Thank you. We replaced “inappropriate” by “irrelevant” as shown in [L43] of
the revised manuscript. 

L#43: please double check your citation style for [14] based on journal
guidelines

Response: Thank you. The reference style is now made consistent, reference # 14 (in
the revised manuscript) with others.

L#58-61: Besides scholars, your study should be valuable to farmers as well. Please
think about other site-specific recommendation practices and mention it in
Introduction section because there is not only geostatistics for optimizing nutrient
use and improve crop production. Countries like, Nigeria, India, Nepal have used
soil testing mobile van as well. I am not sure about similar approach in Ethiopia,
but it is suggested to visit this link (https://doi.org/10.1080/15427528.2017.1387837) for more
information.

Response: Thank you. We appreciate your valuable recommendation and thank for showing
this important link. We mentioned some site-specific recommendation practices and
included in Introduction section. However, Ethiopia has not yet used mobile van so
far for soil testing like the above-mentioned countries. 

L#111: Is this correct amount? 20-50 mg of dried soil; please cross-check with this
information: 

https://www.environment.nsw.gov.au/resources/soils/testmethods/oc.pdf

Response: Thank you for indicating this useful source. We have added a relevant
source [reference # 45] to get more elaboration on the idea of “the amount of dried
soil required in examining OC [L145 in the revised manuscript].

L#136-139: Why there is not any information about tools of spatial interpolation in
Introduction section? You need to re-structure your Introduction and Methodology
(statistical analysis) sections.

No justification- why did you select IDW only, why not others?

Results: there is repetition of information (results) in table and text many times.
Also there is no any value of Table 8, just include mean and SE, and then lsd
comparison. If you have this already, then delete Table 8.

Response: Thank you for the suggestion. We share with your concerns. We tried to
incorporate information about tools of spatial interpolation in Introduction
section. We also re-structured the Introduction and Methodology (statistical
analysis) sections. The justification of IDW selection was included in Introduction
section and Methodology section. The repetition of information (results) in table
and text were minimized and Table 8 was removed.

L#251-252: How GHG information relates to your current study? Do not bring larger
picture in your Discussion section. Please focus on that what you have done and its
implications.

Response: Thank you for the concern. We added some relevant ideas of GHGs and its
implications in Abstract, Introduction, Discussion, and Conclusion sections.

L#255: what is best land planning? 

Response: Thank you for the comment. We replaced “relevant” instead of “best” in the
phrase “best land planning” (L296). A relevant land-use planning gives time and
resources to decision-making processes in order to reach conclusions on suitable or
best possible use of land based on long-term objectives and benefits that are more
equitable.

Reviewer #2: 

The authors presented a work on "Spatial variability of soil properties under
different land-uses in Northwest Ethiopia" but however dwelt so much on land use
instead of the spatial variability. The authors use only 60 samples to produce the
map of the distribution of soil chemical properties in the study location of more
than 6,000 ha. This to me is not acceptable because the samples are too few to
highlight the spatial variability in the soil properties. Even the map produced was
not mentioned in the abstract let alone discussing them in the results and
Discussion section. A number of information that is supposed to be in the abstract
are missing. There were a lot of long sentences that need to be summarized or broken
into two or more sentences.

Response: Thank you for the concern. In considering the soil variability, composite
and purposive soil sampling was employed to reduce sampling intensity. Moreover,
most coverage of the study area is gentle slope, bordering Lake Tana. Supportive
references were also cited [43-44].

a soil sampler per 100 ha was used in India [#reference 43] in revised
manuscript

a soil sample per 625 ha was taken in Turkey [#reference 44] in revised
manuscript

We included the soil variability and soil maps in the Abstract, Methodology, Results,
discussion, and conclusion sections and long sentences were also shortened. 

6. PLOS authors have the option to publish the peer review history of their article
(what does this mean?). If published, this will include your full peer review and
any attached files.

If you choose “no”, your identity will remain anonymous but your review may still be
made public.

Do you want your identity to be public for this peer review? For information about
this choice, including consent withdrawal, please see our Privacy Policy.

Reviewer #1: No

Reviewer #2: No

Response: Thank you. We have used PACE with this submission, so this should be
right.

Please note that once again, thank you very much. Your comments are greatly
appreciated.

Best regards,

Gizachew Ayalew Tiruneh (on behalf of all co-authors)

Lecturer in Debre Tabor University

Ph.D. Fellow in soil science, Bahir Dar University

Email: tiruneh1972@gmail.com

---

## [Decision Letter · Decision Letter 1]

11 May 2021

PONE-D-21-03433R1

Spatial variability of soil chemical properties under different land-uses in
Northwest Ethiopia

PLOS ONE

Dear Dr. Tiruneh,

Thank you for submitting your manuscript to PLOS ONE. After careful consideration, we
feel that it has merit but does not fully meet PLOS ONE’s publication criteria as it
currently stands. Therefore, we invite you to submit a revised version of the
manuscript that addresses the points raised during the review process.

Please submit your revised manuscript by Jun 25 2021 11:59PM. If you will need more
time than this to complete your revisions, please reply to this message or contact
the journal office at plosone@plos.org. When
you're ready to submit your revision, log on to https://www.editorialmanager.com/pone/ and select the 'Submissions
Needing Revision' folder to locate your manuscript file.

If you would like to make changes to your financial disclosure, please include your
updated statement in your cover letter. Guidelines for resubmitting your figure
files are available below the reviewer comments at the end of this letter.

We look forward to receiving your revised manuscript.

Kind regards,

Remigio Paradelo Núñez

Academic Editor

PLOS ONE

Journal Requirements:

Additional Editor Comments (if provided):

One of the reviewers has made several suggestions to improve the manuscript (please
see attached file), in particular the conclusion section needs to be rewritten. In
addition, English grammar should be carefully revised and improved.

Reviewers' comments:

Reviewer's Responses to Questions

**Comments to the Author**

1. If the authors have adequately addressed your comments raised in a previous round
of review and you feel that this manuscript is now acceptable for publication, you
may indicate that here to bypass the “Comments to the Author” section, enter your
conflict of interest statement in the “Confidential to Editor” section, and submit
your "Accept" recommendation.

Reviewer #1: All comments have been addressed

Reviewer #2: (No Response)

2. Is the manuscript technically sound, and do the data
support the conclusions?

Reviewer #1: Yes

Reviewer #2: Partly

3. Has the statistical analysis been performed
appropriately and rigorously? 

Reviewer #1: Yes

Reviewer #2: Yes

4. Have the authors made all data underlying the
findings in their manuscript fully available?

Reviewer #1: No

Reviewer #2: No

5. Is the manuscript presented in an intelligible
fashion and written in standard English?

Reviewer #1: Yes

Reviewer #2: No

6. Review Comments to the Author

Reviewer #1: Dear author(s),

Thank you very much for addressing my concerns.

It was a pleasure to work on your manuscript and provide some constructive
suggestions/ comments.

Reviewer #2: It is true that the authors have tried to address some of the concerns
raised in the previous version of the manuscript. However, the authors have failed
to present their results in a clear and concise manner. Some of the sentences are
not clear and many are laden with lots of grammatical errors. The authors are also
advised to write their conclusion in such a manner that it represents the result of
their study

7. PLOS authors have the option to publish the peer
review history of their article (what does this mean?). If published, this will
include your full peer review and any attached files.

If you choose “no”, your identity will remain anonymous but your review may still be
made public.

**Do you want your identity to be public for this peer review?** For
information about this choice, including consent withdrawal, please see our
Privacy Policy.

Reviewer #1: No

Reviewer #2: No

---

## [Author Response · Author response to Decision Letter 1]

27 May 2021

Date: May 23, 2021

Rebuttal letter

PONE-D-21-03433

We are glad about the academic editor and the reviewers’ comments, which strengthen
the current version of the manuscript “Spatial variability of soil properties under
different land-uses in Northwest Ethiopia”. In addition, our utmost sincere
gratitude goes to you and the reviewers who devote their valuable time to bring our
manuscript to a qualified paper. 

We have provided a one by one reply to all concerns and comments given below. We
thank you for your consideration of this resubmission and look forward to your
response.

Best regards,

Gizachew Ayalew Tiruneh (on behalf of all co-authors)

Lecturer in Debre Tabor University

Ph.D. Fellow in soil science, Bahir Dar University

Email: tiruneh1972@gmail.com

 

Editor’s comments

Comments 1: A rebuttal letter that responds to each point raised by the academic
editor and reviewer(s). You should upload this letter as a separate file labeled
'Response to Reviewers'.

Response: We addressed the concerns provided by the editor and reviewers and uploaded
a file labeled “Response to Reviewers”.

Comments 2: A marked-up copy of your manuscript that highlights changes made to the
original version. You should upload this as a separate file labeled 'Revised
Manuscript with Track Changes'.

Response: We tried to do it.

Comments 3: An unmarked version of your revised paper without tracked changes. You
should upload this as a separate file labeled 'Manuscript'.

Response: We have addressed accordingly.

Comments 4: If you would like to make changes to your financial disclosure, please
include your updated statement in your cover letter. 

Response: We have not made any changes to financial disclosure.

Comments 5: Guidelines for resubmitting your figure files are available below the
reviewer comments at the end of this letter.

Response: We made corrections as per the guidelines 

Comments 6: If applicable, we recommend that you deposit your laboratory protocols in
protocols.io to enhance the reproducibility of your results. Protocols.io assigns
your protocol its own identifier (DOI) so that it can be cited independently in the
future. For instructions see: http://journals.plos.org/plosone/s/submission-guidelines#loc-laboratory-protocols.
Additionally, PLOS ONE offers an option for publishing peer-reviewed Lab Protocol
articles, which describe protocols hosted on protocols.io. Read more information on
sharing protocols at https://plos.org/protocols?utm_medium=editorial-email&utm_source=authorletters&utm_campaign=protocols.

Response: We do not have our own publishable laboratory protocols 

 

Comments 8: Please review your reference list to ensure that it is complete and
correct. If you have cited papers that have been retracted, please include the
rationale for doing so in the manuscript text, or remove these references and
replace them with relevant current references. Any changes to the reference list
should be mentioned in the rebuttal letter that accompanies your revised manuscript.
If you need to cite a retracted article, indicate the article’s retracted status in
the References list and also include a citation and full reference for the
retraction notice.

Response: We have reviewed and checked that the references are complete and
correct.

Reviewers' comments:

Reviewer #1: 

Comments 1. If the authors have adequately addressed your comments raised in a
previous round of review and you feel that this manuscript is now acceptable for
publication, you may indicate that here to bypass the “Comments to the Author”
section, enter your conflict of interest statement in the “Confidential to Editor”
section, and submit your "Accept" recommendation.

Reviewer #1: All comments have been addressed

Reviewer #2: (No Response)

Response: Dear Reviewers, thank you so much for taking your valuable time to elevate
the quality of our manuscript. 

2. Is the manuscript technically sound, and do the data support the conclusions?

Reviewer #1: Yes

Reviewer #2: Partly

Response: Thank you. We have gone thoroughly the revised manuscript, and hopefully
that the second Reviewer will be satisfied.

3. Has the statistical analysis been performed appropriately and rigorously?

Reviewer #1: Yes

Reviewer #2: Yes

Response: Thank you.

4. Have the authors made all data underlying the findings in their manuscript fully
available?

Reviewer #1: No

Reviewer #2: No

Response: Thank you. We have described the data in the manuscript and attached as
supporting information (S1 table and S2 table), and we do hope that the Reviewers’
concerns will be addressed.

5. Is the manuscript presented in an intelligible fashion and written in standard
English?

Reviewer #1: Yes

Reviewer #2: No

Response: Thank you. We have thoroughly revised our manuscript with the help of
Grammarly (premium) and Turnitin software, and we do hope that the second Reviewer’s
concerns will be addressed.

6. Review Comments to the Author

Reviewer #1: Dear author(s),

Thank you very much for addressing my concerns.

It was a pleasure to work on your manuscript and provide some constructive
suggestions/ comments.

Reviewer #2: It is true that the authors have tried to address some of the concerns
raised in the previous version of the manuscript. However, the authors have failed
to present their results in a clear and concise manner. Some of the sentences are
not clear and many are laden with lots of grammatical errors. The authors are also
advised to write their conclusion in such a manner that it represents the result of
their study

Response: Thank you. We have revised the results and conclusions and we do hope that
the second Reviewer’s concerns will be addressed. .

7. PLOS authors have the option to publish the peer review history of their article
(what does this mean?). If published, this will include your full peer review and
any attached files.

If you choose “no”, your identity will remain anonymous but your review may still be
made public.

Do you want your identity to be public for this peer review? For information about
this choice, including consent withdrawal, please see our Privacy Policy.

Reviewer #1: No

Reviewer #2: No

Response: Yes

Response: Thank you. We have used PACE with this submission, so this should be
right.

Please note that once again, thank you very much. Your comments are greatly
appreciated.

Best regards,

Gizachew Ayalew Tiruneh (on behalf of all co-authors)

Lecturer in Debre Tabor University

Ph.D. Fellow in soil science, Bahir Dar University

Email: tiruneh1972@gmail.com

to Reviewers.docx
---

## [Editor Report · Decision Letter 2]

31 May 2021

Spatial variability of soil chemical properties under different land-uses in
Northwest Ethiopia

PONE-D-21-03433R2

Dear Dr. Tiruneh,

We’re pleased to inform you that your manuscript has been judged scientifically
suitable for publication and will be formally accepted for publication once it meets
all outstanding technical requirements.

Kind regards,

Remigio Paradelo Núñez

Academic Editor

PLOS ONE
---

## [Editor Report · Acceptance letter]

7 Jun 2021

PONE-D-21-03433R2 

Spatial variability of soil chemical properties under different land-uses in
Northwest Ethiopia 

Dear Dr. Tiruneh:

I'm pleased to inform you that your manuscript has been deemed suitable for
publication in PLOS ONE. Congratulations! Your manuscript is now with our production
department. 

Kind regards, 

on behalf of

Dr. Remigio Paradelo Núñez 

Academic Editor

PLOS ONE